# Exploring Physical Characterization and Different Bio-Applications of *Elaeagnus angustifolia* Orchestrated Nickel Oxide Nanoparticles

**DOI:** 10.3390/molecules28020654

**Published:** 2023-01-09

**Authors:** Banzeer Ahsan Abbasi, Javed Iqbal, Tabassum Yaseen, Syeda Anber Zahra, Saima Ali, Siraj Uddin, Tariq Mahmood, Sobia Kanwal, Hamed A. El-Serehy, Wadie Chalgham

**Affiliations:** 1Department of Botany, Rawalpindi Women University, 6th Road, Satellite Town, Rawalpindi 46300, Pakistan; 2Department of Botany, Bacha Khan University, Khyber Pakhtunkhwa, Charsadda 24540, Pakistan; 3Department of Plant Sciences, Quaid-i-Azam University, Islamabad 45320, Pakistan; 4Department of Biochemistry, Quaid-i-Azam University, Islamabad 45320, Pakistan; 5Department of Biology and Environmental Sciences, Allama Iqbal Open University, Islamabad 44000, Pakistan; 6Department of Zoology, College of Science, King Saud University, Riyadh I1451, Saudi Arabia; 7Department of Mechanical and Aerospace Engineering, University of California, Los Angeles, CA 90095, USA

**Keywords:** green synthesis, nickel oxide nanoparticles, characterization, bio-applications

## Abstract

*Elaeagnus angustifolia* (EA) mediated green chemistry route was used for the biofabrication of NiONPs without the provision of additional surfactants and capping agents. The formation of NiONPs was confirmed using advanced different characterization techniques such as Scanning electron microscopy, UV, Fourier transmission-infrared, RAMAN, and energy dispersal spectroscopic and dynamic light scattering techniques. Further, different biological activities of EA-NiONPs were studied. Antibacterial activities were performed using five different bacterial strains using disc-diffusion assays and have shown significant results as compared to standard Oxytetracycline discs. Further, NiONPs exhibited excellent antifungal performance against different pathogenic fungal strains. The biocompatibility test was performed using human RBCs, which further confirmed that NiONPs are more biocompatible at the concentration of 7.51–31.25 µg/mL. The antioxidant activities of NiONPs were investigated using DPPH free radical scavenging assay. The NiONPs were demonstrated to have much better antioxidant potentials in terms of % DPPH scavenging (93.5%) and total antioxidant capacity (81%). Anticancer activity was also performed using HUH7 and HEP-G2 cancer cell lines and has shown significant potential with IC_50_ values of 18.45 μg/mL and 14.84 μg/mL, respectively. Further, the NiONPs were evaluated against *Lesihmania tropica* parasites and have shown strong antileishmanial potentials. The EA-NiONPs also showed excellent enzyme inhibition activities; protein kinase (19.4 mm) and alpha-amylase (51%). In conclusion, NiONPs have shown significant results against different biological assays. In the future, we suggest various in vivo activities for EA-NiONPs using different animal models to further unveil the biological and biomedical potentials.

## 1. Introduction

Nanobiotechnology is an emerging area that involves the fabrication of nanoparticles utilizing bacteria, algae, fungi, and plants ranging from 1–100 nm in size (i.e., one billionth part of a meter) [1,2]. This fast-growing field has versatile applications at different commercial levels, i.e., agriculture, cosmetics, catalysis, food, bio-remediation, optics, cancer theranostics, drug delivery, material engineering, and many more [3,4,5]. Metal oxide nanoparticles (MONPs) that are fabricated via the green route have gained much attention due to their safe, benign, energy-saving, cost-effective, environment-friendly, and reproducible nature [6,7]. The nano-sized particles have attracted the attention of the scientific/research community for being variable from their macro/bulk counterparts and because of their fascinating physical, optical, mechanical, magnetic, sensing, and electrical properties [8,9,10]. Up till now, Different nanoparticles, including various metal oxides such as cobalt oxide (CO_2_O_3_) [11], nickel oxides (NiO) [12], silver oxides (Ag_2_O, AgO) [13], iron oxides (FeO, Fe_2_O_3_, Fe_3_O_2_) [14], copper oxide (CuO) [15], palladium oxide (PdO) and zinc oxide (ZnO) [16], etc. with multi-functional properties (due to their size, nature, shape, charge and the method they are synthesized) have been orchestrated. Among different MONPs, Nickel-oxide nanoparticles (NiONPs) have gained much attention from the research community due to their remarkable intrinsic properties such as chemical stability, p-type semi conductance, nano-devices fabrication, electron transferring potentials, sensing, electro-catalysis, optoelectronic abilities, super capacities, and drug delivery. So far, NiONPs have been investigated for their anti-inflammatory, cytotoxic, and pollutant adsorption properties [12,17].

Previously, different physico-chemical routes, including galvanostatic anodization, electro-deposition, sol-gel chemistry, combustion, hydrothermal synthesis, solvothermal method, etc., have been utilized for the orchestration of NiONPs [18,19,20,21]. However, NPs orchestrated via these routes have serious limits in biomedical applications owing to their eco-toxicity, massive energy, temperature, and pressure inputs, and production of their non-biodegradable products [22,23]. In order to cope with these challenges, a reliable, quick, eco-friendly, reproducible, and greener method was unraveled. Phytosynthesis of nanosized particles became popular as medicinal plants possess green chemicals alkaloids, tannins, flavonoids, terpenoids, saponins, etc. [24,25]. These green chemicals are considered effective reducing, capping, chelating, and stabilizing agents. Therefore, the orchestration of nanosized particles utilizing different plants is under serious consideration [26].

Recently, NiONPs have been fabricated via phytochemicals and are investigated for different biological applications. It is to be noted that many plants, viz. *Agathosma betulina* [27], *Gerinium wallichianum* [28], *Bergenia ciliate* [29], *Rhamnus virgata* [3], *Nephelium lappaceum* [30], *Berberis balochistanica* [4,31] and *Tamarix serotine* [32], have been explored for the synthesis of NiONPs. One such naturally occurring plant having important pharmacological and medicinal values is *Elaeagnus angustifolia* (family Elaengnaceae, local name Sangzala) owing to the presence of different green chemicals vitamins (vitamin B1, tocopherol, α-carotene, and vitamin C), flavonoids, sugars, sterols, alkaloids, sugar, minerals, protocatechuic, p-hydroxybenzoic and caffeic acid, quercetin, isorhamnetin and kaempferol [33,34]. *E. angustifolia* (ElA) is reported to alleviate several serious ailments such as jaundice, cancer, asthma, inflammation, osteoarthritis, diarrhea, gastrointestinal problems, and rheumatoid arthritis. According to ethnobotanical data, different ElA extracts have shown potential antiulcer, antimicrobial, anti-inflammatory, antioxidant, and wound-healing abilities [33,35]. In this study, EIA-based NiONPs were characterized and screened for anticancer, antileishmanial, antimicrobial, antioxidant, enzyme inhibition potencies, and biocompatibility assay.

## 2. Results

### 2.1. Characterization of ElA-NiONPs

The *E. angustifolia*-based NiONPs were screened via different spectroscopic (Raman, EDS, DLS, UV, FTIR, XRD) and microscopic techniques (SEM). During the assessment of vibrational properties via Raman, active vibrational modes were noted at 375.92cm^−1^ (1P), 568.79cm^−1^ (1P), 704.44cm^−1^ (2P), 1059.86^−1^ (2P) and 1680.67cm^−1^ (1M). The results for Raman are illustrated in Figure 1A. For the analysis of purity and atomic contents of synthesized ElA-NiONPs, EDX was performed. The elemental analysis depicted in Figure 1B has shown the presence of nickel and oxygen. The presence of impurity (any other element) was missing, which clearly indicated the single-crystalline nature of NiONPs. For the measurement of hydrodynamic size, an index of polydispersity and ζ-potentials DLS was performed. The results revealed huge-sized nanoparticles aggregates of ~21 nm. The index of polydispersity and zeta potential of IONPs was 1 with a standard deviation of 108 d. nm and −11 mV, respectively Figure 2A,B.

Furthermore, the UV vis-absorption spectrum was obtained utilizing a spectrometer (UV–Vis spectrophotometer (Munich, Germany), and the UV absorption peak was noted at 332 nm (Figure 3A). In the FTIR spectrum of *E. angustifolia* mediated NiONPs, peaks present at 3278.11 cm^−1^, 1614.13 cm^−1^, 1417.46 cm^−1^, 995.99 cm^−1^ signify O-H, C=C, -C=O and C-H stretching. However, peaks at 545.45 and 681.77 cm^−1^ correspond to NiO bond vibrations from NiONPs. The results of the FTIR spectrum are illustrated in Figure 3B. Further, the XRD data revealed different diffraction peaks with 2 theta values of 26.98 (100), 36.84 (111), 43.47 (200), 61.88 (220), 75.75 (311), 79.38 (222), indicating impurities-free ElA-NiONPs reflecting the face-centered cubic phase structure of the NiO phase with a lattice parameter of 4.17710 Å, which is in accordance with the standard card of JCPDS NO. 47–1049. No additional peaks were observed in the XRD patterns, indicating the purity of the sample. The average size calculated via Origin-pro was ~21 nm Figure 3C. The crystallinity of the ElA-NiONPs is 14.00616399, calculated via Origin-pro. Moreover, SEM depicted in Figure 4 showed spherical NPs.

### 2.2. Enzyme Inhibitory Assay

The PK enzymes are critical in anticancer studies as these enzymes are responsible for adding phosphate to threonine-serine and tyrosine residues. These amino acids have a significant role in cellular differentiation and apoptosis. The improper functioning of these enzymes results in the progression and growth of tumor cells. The *E. angustifolia* mediated NiONPs were tested for their PK inhibitory activity. The ElA-NiONPs were found to have a maximum inhibition zone of 19.4 mm at the highest NP concentration of 500 μg/mL and as shown in Figure 5A.

To verify the antidiabetic ability of ElA-NiONPs, alphaamylase activity (AA) was investigated, as these enzymes are important during the conversion of carbohydrates to simple molecules. Their deficiency and improper functioning are associated with glucose exertion and diabetes. Therefore, particular entities having inhibitory action against AA enzymes are of noteworthy interest in diabetes research. The *E. angustifolia*-mediated NiONPs have shown significant antidiabetic activity and yielded maximum inhibition of 51% at a concentration of 500 µg/mL. The Acarbose inhibited AA-enzymes up to 92%. However, at lower concentrations, these ElA-NiONPs were found to be ineffective, i.e., <31.3 µg/mL, as illustrated in Figure 5B.

### 2.3. Anticancer Activity of ElA-NiONPs

Cancer is a prevalent disease with a high mortality rate around the globe. The scientific community is striking hard to unravel novel drugs with significant potential against the cancerous cell. In this perspective, ElA-based NiONPs were explored against HUH7 and HEP-G2 for their anti-cancerous activities. These cancerous cells were treated with ElA-NiONPs (dose range: 3.91–500 µg/mL) for 24 h. The asynthesized NiONPs have shown excellent potential against selected cancerous cells. The mortality percentage observed was 82% and 80% for HUH7 and HepG2 cells, respectively. The greenly synthesized nickel nano-size particles have shown concentration-dependent responses. Figure 5C indicated IC_50_ values of 18.45 μg/mL and 14.84 μg/mL for HUH7 and HepG2 cells, respectively.

### 2.4. Antileishmanial Activity of ElA-NiONPs

*L. tropica* is responsible for causing an infectious disease, leishmaniasis. Leishmaniasis is an NTD (neglected tropical infection) mostly spread by sandflies-bites. This parasite became resistant to antimonials, and other drugs formulated were used to cure this parasite. The research community is continuously working on the formulation of some effective alternate drugs. Therefore, in this study, different concentrations (range: 3.91–500 µg/mL) of ElA-NiONPs were tested against amastigotes and promastigotes of *L. tropica.* The ElA-NiONPs have shown significant potentials with IC_50_ values; 33.75 μg/mL for promastigotes and 54.15 μg/mL for amastigotes, as shown in Figure 5D.

### 2.5. In Vitro Antifungal and Antibacterial Assay

Greenly prepared NiONPs were evaluated against five fungal strains. Figure 6A presents the anti-fungus activity of ElA-NiONPs across the dose range of 15.63–500 µg/mL. The Agar-diffusion tests were exploited in the concentrations range of 15.63–500 µg/mL. Our test results confirmed the highest susceptibility of MRa (MIC: 15.63 µg/mL) whilst lower susceptibility of Afl (MIC: 62.5 µg/mL). The MIC value for other fungal stains (ANi, Cai, and FSo) was 31.25 µg/mL. Also, antibacterial assays were performed against five different infectious strains of bacteria across different doses (15.63–500 µg/mL) with oxytetracycline as a positive control. All bacterial strains used were observed to be inhibited across all tested doses indicating ElA-NiO as a potential antibacterial agent. The MIC values of 15.3 µg/mL, 62.5 µg/mL, 15.3 µg/mL, 31.3 µg/mL, and 31.3 µg/mL were recorded for SAu, Kpn, BSu, Pae, and Eco, respectively. From Figure 6B, it can be confirmed that SAu (MIC: 15.3 µg/mL) was found to be the least susceptible, while the highest susceptibility was noted for KPn.

### 2.6. In Vitro Biocompatibility Test

The % hemoglobin rupture and related biocompatible nature of ElA-NiONPs were investigated by performing a hemolytic test against human red cells (Figure 6C). The highest hemoglobin rupturing rate was observed at the highest dose, i.e., 500 µg/mL, whilst no hemolysis was noted on lower doses confirming the biosafe nature of asynthesized nanosize particles. The % hemolysis observed for tritonX100 was 74%.

## 3. Discussion

A complete, robust, and green method has been revealed for the fabrication of nanosized (21 nm) particles utilizing green chemicals of *E. angustifolia*. The well-documented medicinal uses of *E. angustifolia* are due to the presence of vitamins, flavonoids, sugars, sterols, alkaloids, sugar, minerals, protocatechuic, p-hydroxybenzoic, etc. These active compounds might have an important role as reducing and stabilizing agents. Single phase, nanosized (21 nm), spherical shape, and pure *E. angustifolia*-mediated NiONPs with negative zeta potential were successfully demonstrated through numerous characterization techniques. Several functional groups (Carbonyl, hydroxyl, alkenes, alkyl, and aromatic compounds) confirmed via FTIR represent the green chemicals that have active participation against various ailments [33,34]. Previous studies also reported the fabrication of NiONPs synthesis via green methods [28,36,37]. The presence of the main bands was identified at 545 cm^−1^ and 681 cm^−1^ and ascribed to the stretching mode associated with Ni-O vibration. Zhang et al. [38] also synthesized green nanoparticles using *Eichhornia crassipes* (Ec) extract and got similar results. The band at 3278 cm^− 1^ corresponds to the vibration (stretching mode of H-O-H) of absorbed water molecules kannan et al., 2020 [39]. In general, the FTIR spectrum was in good agreement with the XRD data, which confirmed the fact that this synthesis produced high-quality NiONP. In RAMAN active mode pattern, the scattering peak appearing at 375.92 cm^−1^, and 704.44 cm^−1^ mode are ascribed to the symmetrical stretching modes of NiO NPs due to the inelastic scattering of light [40]. The peak at ~568.79 cm^−1^ was directly associated with the contribution of one-phonon (1P) transverse optical (TO) and longitudinal optical (LO) modes. The other band at ~1059.86 cm^−1^ is attributed to + LO mode. Further, 1680.67 (2M) is one of the more intense in the NiO Raman spectrum [41]. Excellent PK and AA inhibition potencies were recorded, which reflect that our ElA-NiONPs might have significant potential against diabetes and cancer, which is in agreement with the previous reports [30,39,40]. Also, the potential cytotoxic nature of *E. angustifolia* orchestrated NiONPs was confirmed against cancer cells (HUH7, HEPG2) and Leishmania (amastigotes, promastigotes). Previously, Abbasi et al. [41] and Iqbal et al. [42] have also reported the concentration-dependent mortality of cancerous and leishmanial cells. Further, the bioinspired NiONPs have shown significant MICs against various fungus and bacterial strains utilized in this study. All the concentrations of nano-sized particles were found effective/active against the exploited microbial strains. It is considered that these potentials are due to interferences of nanosized particles with intracellular machinery, causing the leakage of the plasma membrane [7,43]. These results are congruent with the previous studies [31,44,45,46], where *Berberis balochistanica* leaf and stem-based NiONPs were found to inhibit the growth of various antimicrobial strains. Furthermore, % hemoglobin release was investigated via hemolytic assay. Negligible hemoglobin rupturing (hemolysis) at lower concentrations indicated the biosafe nature of ElA-mediated NiONPs. Our results are similar to some previous studies of Iqbal et al. [46], Zhang et al. [47], Iqbal et al. [48], and Hameed et al. [49], where they also have synthesized biosafe *Rhamnus virgata*, *Myristica fragrans*, and *Rhamnus triquetra* based nanosized particles.

## 4. Materials and Methods

### 4.1. Sampling and Processing

The leaves of *Elaeagnus angustifolia* were collected from the upper mountainous area of Parachinar, Pakistan, and were taxonomically recognized and confirmed by Dr. Afzal Shah from Quaid-i-Azam university Islamabad. The ElA leaves were shade-dried for ∼1 month to remove water content. The obtained dry leaves were grounded to a fine powder. Further, 15-g powder of *E. angustifolia* was suspended in 200 mL of nanopure-water while stirring continuously at 85 °C for ∼1.5 h. The resultant solution was cooled and filtered three times utilizing Whatman papers. This final yellow color aqueous extract was used for the fabrication of nano-sized particles.

### 4.2. Fabrication of the Targeted ElA-NiONPs

The NiONPs were fabricated by adding Nickel nitrate (1.5 g) into 50 mL of filtered *E. angustifolia* aqua leaves extract. The reaction mixture was continuously stirred and heated on a hot plate at ∼75 °C for 1.5 h for maximum reactions. The color change from yellow to greenish black was observed for plant extract. The resultant suspension was cooled for ∼0.5 h and centrifuged at 4000 rpm/30 min. The pellet settled at the bottom, comprised of NiONPs, was washed and centrifuged four times using nanopure-water for the removal of Ni ions and green residues. The pellet was incubated at 100 °C/2 h, and powder of ElA-NiONPs was obtained. This powder (considered as NiONPs) was characterized for different microscopic and spectral studies and screened for multiple biological applications.

### 4.3. Characterization of ElA-NiONPs

The EIA-NiONPs were characterized through Raman, EDS, DLS, UV, SEM, FTIR, and XRD). The vibrational properties through Raman spectral studies of ElA-NiONPs were investigated by noticing active vibrational modes using Thermoscientific Instrument (Waltham, Massachusetts, U.S.) on laser excitation of 514 nm. For analyzing the exact atomic contents and their associated purities, EDX was performed for ElA-NiONPs. Further, 200 μg of ElA-NiONPs were suspended in 200 mL nanopure water by bath sonicator for forty min. This suspension was transferred to a capillary cell for the measurement of conductivity, hydrodynamic size, index of polydispersity, and potential measurements using Malvent Zetasizer (Almelo, The Netherlands). The reduction rate and progression of fabrication of NPs were investigated utilizing UV–vis spectrophotometry (Shimadzu 2450; Duisburg, Germany). The spectrum background was corrected and adjusted against DMSO. The morphology and surface structure were studied using SEM (Scanning electron microscope (SEM-EDS (Evo MA10-ZEISS; Oberkochen, Germany) Instrument). The transmission FTIR spectra for ElA-NiONPs was obtained using FTIR-spectrometer (Alpha, Bruker, Germany) viz KBr pellet procedure (range: 500–5000 cm^–1^) for the investigation of the different biomolecules present upon the surface of synthesized particles. The transmission spectrum was analyzed via Origin-Pro-8.5. For this aim, samples were blended with the KBr powder and converted into the pellet by pressing it; the pellet was subjected to FTIR spectroscopic analysis in the spectral range of 4000 to 500 cm^−1^, with the 4 cm^−1^ resolution, as explained by Sharifi-Rad and Pohl [50]. Furthermore, the purity and approximate crystalline size was also recorded using an x-rays diffractometer (model Bruker AXS D8 Advance; New York, NY, USA) supplied with the source of copper radiations Cu-Kα radiation (λ = 1.54 Å) with 45 kV and 30 mA voltage and current, respectively.

### 4.4. Sample Preparations and Bio-Applications of ElA-NiONPs

The suspension of 10 mg ElA-NiONPs in 99% DMSO was prepared for screening different bio-applications in vitro.

#### 4.4.1. Protein Kinase Assay

The PK assay was determined via previous methods (Faisal et al., 2021a). For examining PK inhibitions potencies, uniform lawns of Streptomyces85E bacterial strain were obtained by adding inoculum (100 μL) to ISP4 media. The filter disc (sterilized) ~6 mm impregnated with 10 mL ElA-NiONPs was carefully kept upon media with bacterial smooth/uniform lawns and incubated for 72 h/30 °C. Different bald zones and clear zones were observed around the filter discs. The appearance of zones reflected the inhibitory effects of ElA-NiONPs against mycelia and spores of the targeted strain. Inhibitory zones were recorded in mm via Vernier caliper. The surfactin was taken as positive, while DMSO was negative control.

#### 4.4.2. Antidiabetic Potential via Alpha-Amylase Assay

For the determination of antidiabetic potentials, alpha-amylase (25 μL), phosphate-buffer saline (15 μL), 10 μL NiONPs starch solution (40 μL) was added to a microplate, incubated at 50 °C. After incubation for 30min, iodine solution (90 μL) and 1M HCl (20 μL) were also added. The suspension obtained was further incubated for 20 min. The DMSO was served as negative while Acarbose was a positive control. The optical density was noted at 540 nm, MIC was measured while exploiting Graphpad software, and the %inhibition was also calculated.

#### 4.4.3. Anticancer Activity of ElA-NiONPs

HepG2 (RCB1648) cancer cells were utilized for the determination of the anticancer ability of ElA-NiONPs via MTT assay. The cancerous cells were cultivated in DMEM media laden with penicillin (1%), FBS (10%), and streptomycin (1%) and incubated at 37 °C in a carbon dioxide (5%) incubator. The MTT assay was executed in an eliza-plate with various doses of ElA-NiONPs (3.91–500 μg mL^−1^) for ∼48 h. With this objective, MTT solutions (20 μL) were loaded in wells of a microplate and incubated for ~3 h. The DMSO was substituted with DMEM media (100 μL) and was incubated for an additional 25 min. The extent of formazan production by alive cells was measured at 570 nm in a microplate reader, and IC_50_ values were noted utilizing Graphpad software. Untreated cells served as control, and %inhibition was measured.

#### 4.4.4. Antileishmanial Studies of ElA-NiONPs

The ElA-NiONPs were screened for their antileishmanial potency utilizing MTT assay against KWH23strain of *L. tropica* (promastigote, amastigote). The parasites were cultivated on MI99-media having FBS (10%). Different concentrations of ElA-NiONPs (3.91–500 µg/mL) were tested against *L. tropica*. The suspension is prepared to utilize the standardized culture (100 μL), fresh media (50 μL), and ElA-NiONPs (50 μL). The 96-well plate containing ElA-NiONPs was incubated for 72 h/24 °C in 5% carbon dioxide incubator, and readings were noted at 540 nm in a microplate reader. IC_50_ values and %inhibition were also recorded.

#### 4.4.5. In Vitro Antifungal and Antibacterial Assay

The fungal strains; *C. albican* (CAl), *F. solani* (FSo), *A. niger* (ANi), *A. flavus* (AFl), and *M. racemosus* (MRa) were exploited for examining the antifungal abilities of ElA-NiONPs via disc diffusion method. To attain this objective, NiONPs were loaded (Conc. range 15.63–500 μg mL^−1^) on filter discs and incubated for 25 °C/24 h. The ZOI and MICs were noted after 20 h. The AmpoB was employed as positive while DMSO as a negative control. Similarly, the bacterial strains; *S. aureus* (SAu), *K. pneumonia* (Kpn), *B. subtilis* (BSu), *P. aeruginosa* (PAe), and *E. coli* (ECo) were utilized for investigating antibacterial activity. For this purpose, ElA-NiONPs (Conc. range 15.63–500 μg mL^−1^) were spread on filter discs, placed upon uniform lawns of bacteria, and incubated for 37 °C/24 h. The ZOI and MICs were noted after 20 h. The oxytetracycline served as a positive while DMSO was the negative control

#### 4.4.6. Biocompatibility Assays

The biocompatible nature of ElA-NiONPs was determined by exploiting the red cells of humans. The hemolytic activities of biological substances are standardized; biological substance is hemolytic when hemolytic activity is ˃5%, slightly hemolytic between 2–5%, and non-hemolytic at 2%. Hemolytic refers to red cell rupturing and subsequent release of hemoglobin. Greater hemolytic activity corresponds to high toxicological effects. The human red cells were exposed to ElA-NiONPs (Conc. Range: 7.81–500 µg/mL), and IC_50_, %hemoglobin rupture (biocompatibility), and toxicology rates were confirmed.

## 5. Conclusions

Overall, the present study directs ElA-NiONPs to inhibit various in vitro, microbial, cancerous, and leishmanial cells. Remarkable enzyme inhibitory activities have been reported in our study. The nanosized particles fabricated using the pure green route were biosafe in nature. Owing to the investigated medicinal properties, ElA-based NiONPs should be explored further for in vivo studies and advanced-level bio-applications, imaging, regeneration, etc.

## Figures and Tables

**Figure 1 molecules-28-00654-f001:**
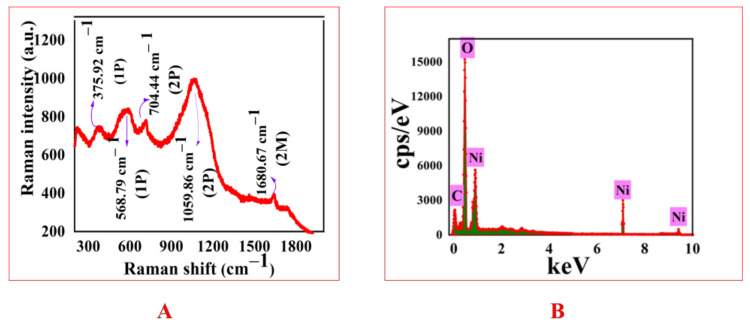
Spectral studies of ElA-NiONPs; (**A**) RAMAN modes; (**B**) EDX analysis.

**Figure 2 molecules-28-00654-f002:**
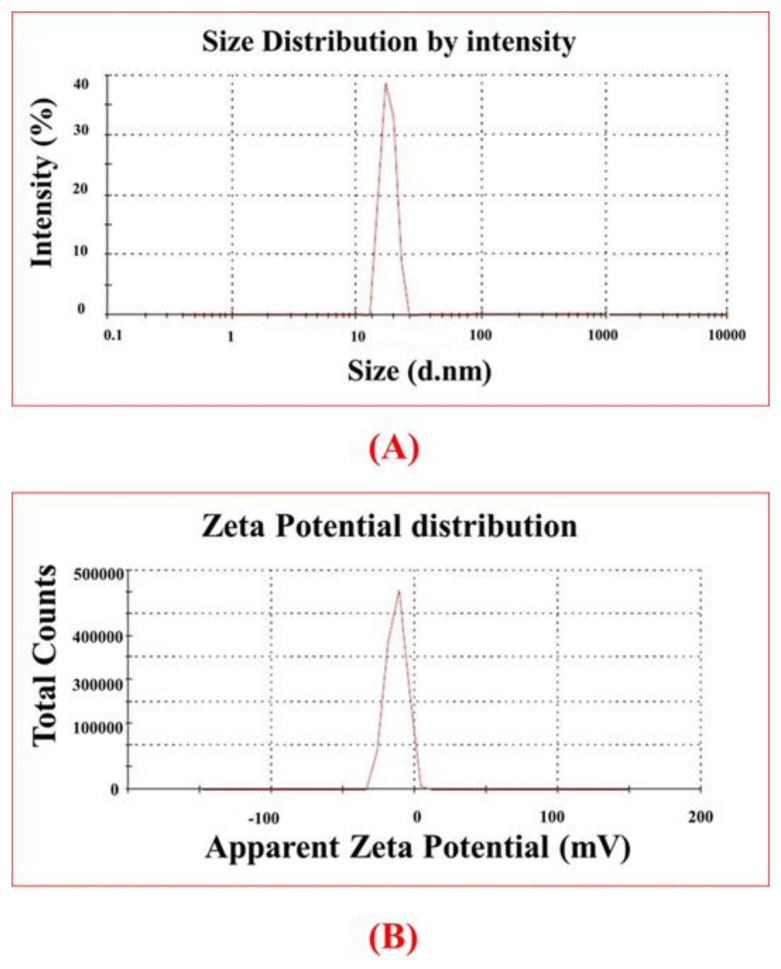
D.L.S spectral studies; (**A**) Zeta size; (**B**) Zeta potential distribution.

**Figure 3 molecules-28-00654-f003:**
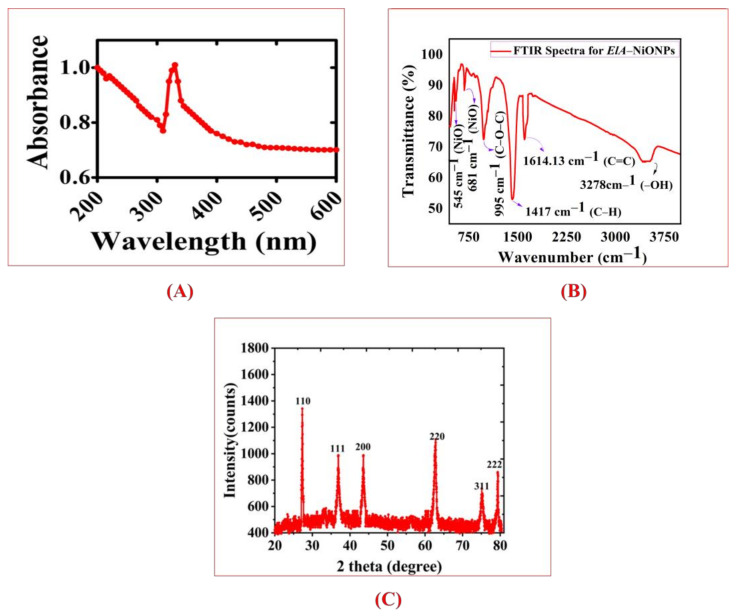
Spectral analysis of *E. angustifolia* mediated NiONPs; (**A**) UV spectra; (**B**) FT-IR transmission spectra; (**C**) XRD studies of ElA-NiONPs.

**Figure 4 molecules-28-00654-f004:**
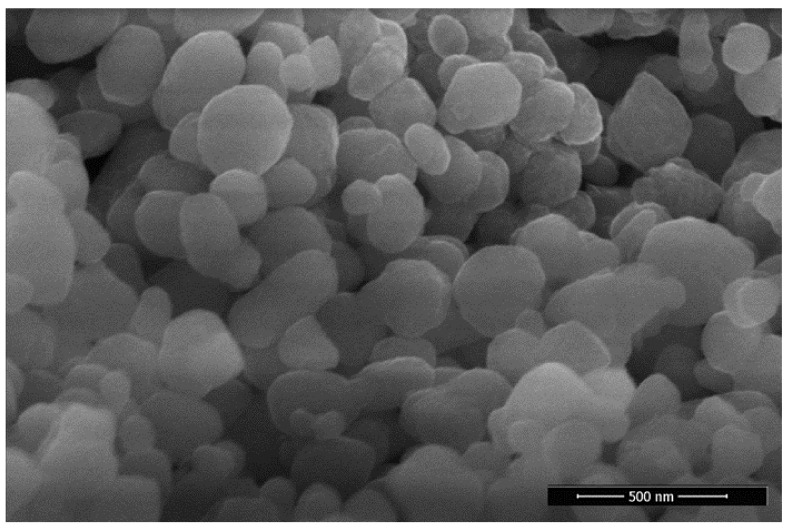
SEM image illustrating NiONPs.

**Figure 5 molecules-28-00654-f005:**
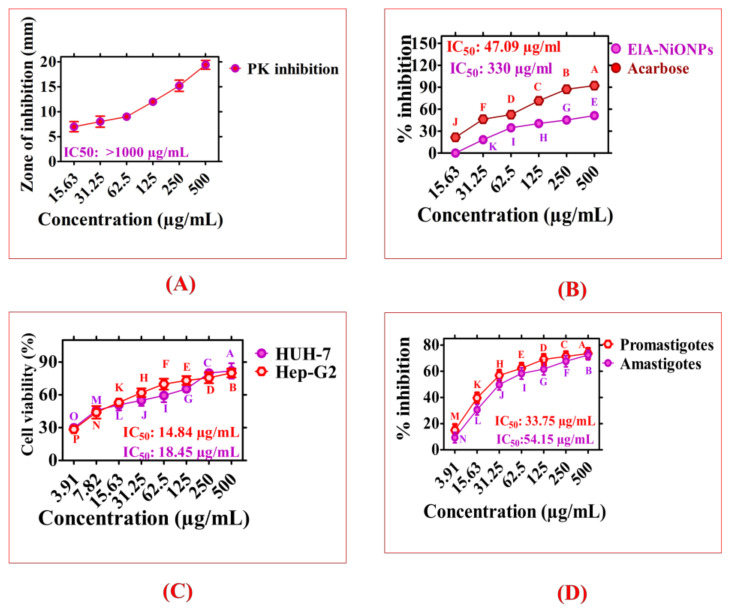
Enzyme inhibition and cytotoxic activities of ElA-NiONPs; (**A**) Protein kinase inhibitory studies; (**B**) Alpha-amylase inhibition; (**C**) Anticancer potentials of ElA-NiONPs; (**D**) Anti-leishmanial studies of ElA-NiONPs. Data represent the mean of three replicates, and each alphabet indicates significance at *p* < 0.05.

**Figure 6 molecules-28-00654-f006:**
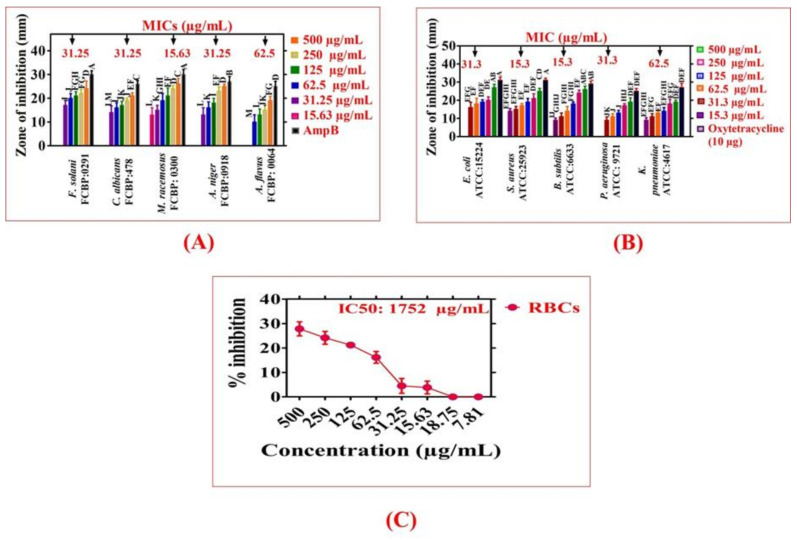
Antimicrobial and Biocompatibility assays for ElA-NiONPs; (**A**) Antifungal assay; (**B**) Antibacterial assay; (**C**) Biocompatibility studies. Data represent the mean of three replicates, and each alphabet indicates significance at *p* < 0.05.

## Data Availability

The data is available from corresponding authors upon request.

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
