# Peer review of "Exploring Physical Characterization and Different Bio-Applications of *Elaeagnus angustifolia* Orchestrated Nickel Oxide Nanoparticles"

_molecules, 2023, doi:10.3390/molecules28020654_

Round 1

Reviewer 1 Report

Major point:

·      Anticancer activity have to be tested also after 48 hours treatments and also on a non-cancerous cell line (i.e. HacaT, HEK293, human fibroblasts or similar) in order to define the specificity of theElA-NiONPs on cancerous cells

Minor points:

·      In 2.5  and 2.6 “invitro” should be  “in vitro”

·      Letters are uppercase in the figures while are lowercase in the captions

·      Statistical analysis should be performed for experiments shown in figure 5 and 6

Author Response

Thank you so much for reviewing our article by providing valuable comments/suggestions. All comments have been properly addressed. Once again the manuscript has been critically reviewed several times for language proficiency, grammar, sentence structures, singularity and plurality of the words. All major and minor mistakes have been removed and corrected. The changes have been highlighted with red color throughout manuscript. If you provide further suggestions/recommendations, we will be very happy to address.

Reviewer 2

Comment: Anticancer activity have to be tested also after 48 hours treatments and also on a non-cancerous cell line (i.e. HacaT, HEK293, human fibroblasts or similar) in order to define the specificity of the ElA-NiONPs on cancerous cells

Thanks a lot for your valuable suggestion. If the purpose of this activity is to check the biocompatibility we are done with biocompatibility assay using RBC. Regretfully, Our lab is not enriched with enough resources to perform further experimental work at this stage on cancer or non-cancerous cell line.

Comment: In 2.5 and 2.6 “invitro” should be  “in vitro”

Response: In vitro has been substituted for invitro as suggested by worthy reviewer.

Comment: Letters are uppercase in the figures while are lowercase in the captions

Response: This comment has been addressed. The targets are height lighted with red colors.

Comment: Statistical analysis should be performed for experiments shown in figure 5 and 6

This comment has been addressed. Statistical analysis has been performed for experiments shown in figure 5 and 6

Reviewer 2 Report

The manuscript “Exploring physical characterization and different bio-applications of Elagnus angustiflora orchestrated nickel oxide nanoparticles” reports about a green route for the synthesis of NiO nanoparticles. Several biological applications are proposed (i.e antibacterial, andtifungal, anticancer). The idea is interesting and several interesting experiments are carried out, but the manuscript is not suitable for publication, in my opinion, in the present form.

Herein some criticisms:

-          In the description of the synthesis procedure: why did the authors choose 1.5 hr at 75°c as first step and not other temperatures or times? Why did the author cool the NPs for 0.5 hr and not more or less and at what temperature (R.T.?)? The washing steps proposed are 4: did the authors stop when the supernatant become transparent or when?

-          What happens to the plant extract during the synthesis procedure? The procedure should be repeated without the Nickel precursor and each step should be characterized to understand the modification on the extract.

-          Raman spectrum of the nanoparticles isn’t explained, at least a reference should be added. Why the 2M mode is absent? It’s usually one of the more intense in NiO Raman spectrum. Further, the Raman spectrum of the plant extract and also of the dried leaves should be added.

-          The FTIR spectrum of the plant extract and also of the dried leaves should be added, this could help understanding the organic moieties present on nanoparticles surface, in order also to better comprehend the biological properties.

-          Where is the TEM the authors refer to?

-          I don’t see agglomeration in the SEM pictures, the nanoparticles seem just bigger than 21 nm.

-          Most important: the same biological tests and assays carried out on the nanoparticles must be performed on the extract in order to be sure monitored properties have to be ascribed to the nanoparticles and not to the extract, which could be very bio-active as well.

-          Discussion section is too brief. All the results should be deeply discussed and comprehended, in order to avoid “a list of properties” without any explanation.

-          All the used instruments (brand and characteristics) must be added in the Materials and Methods section.

-          Sometimes “nickle” is written instead of “nickel”.

-          A curiosity: what do the author mean for nano-purewater?

Author Response

Thank you so much for reviewing our article by providing valuable comments/suggestions. All comments have been properly addressed. Once again the manuscript has been critically reviewed several times for language proficiency, grammar, sentence structures, singularity and plurality of the words. All major and minor mistakes have been removed and corrected. The changes have been highlighted with red color throughout manuscript. If you provide further suggestions/recommendations, we will be very happy to address.

Reviewer 3

Comment: In the description of the synthesis procedure: why did the authors choose 1.5 hr at 75°c as first step and not other temperatures or times? Why did the author cool the NPs for 0.5 hr and not more or less and at what temperature (R.T.?)? The washing steps proposed are 4: did the authors stop when the supernatant become transparent or when?

Response to comment: Thank you so much for your critical review. Highly appreciated. This time and temperature is mentioned in previously published articles as this is optimized time and temperature for synthesis of nanoparticles. For more details and support for the statement provided please see the publication links provided below.

Khalil, A. T., Ovais, M., Ullah, I., Ali, M., Shinwari, Z. K., Hassan, D., & Maaza, M. (2018). Sageretia thea (Osbeck.) modulated biosynthesis of NiO nanoparticles and their in vitro pharmacognostic, antioxidant and cytotoxic potential. Artificial cells, nanomedicine, and biotechnology46(4), 838-852.

Khalil, A. T., Ovais, M., Ullah, I., Ali, M., Shinwari, Z. K., Khamlich, S., & Maaza, M. (2017). Sageretia thea (Osbeck.) mediated synthesis of zinc oxide nanoparticles and its biological applications. Nanomedicine12(15), 1767-1789.\

Iqbal, J., Abbasi, B. A., Mahmood, T., Hameed, S., Munir, A., & Kanwal, S. (2019). Green synthesis and characterizations of Nickel oxide nanoparticles using leaf extract of Rhamnus virgata and their potential biological applications. Applied Organometallic Chemistry33(8), e4950.

Uddin, S., Safdar, L. B., Anwar, S., Iqbal, J., Laila, S., Abbasi, B. A., ... & Quraishi, U. M. (2021). Green synthesis of nickel oxide nanoparticles from Berberis balochistanica stem for investigating bioactivities. Molecules26(6), 1548.

 The washing step was performed until we got clear supernatant.

Comment: What happens to the plant extract during the synthesis procedure? The procedure should be repeated without the Nickel precursor and each step should be characterized to understand the modification on the extract.

Response to comment: Thank you so much for your revision. We have provided detailed mechanisms of plant extract preparation mechanisms in material and methods section. Please see Fabrication of the targated ElA-NiONPs (4.2.)

Comment: Raman spectrum of the nanoparticles is n’t explained, at least a reference should be added. Why the 2M mode is absent? It’s usually one of the more intense in NiO Raman spectrum. 

Response to comment: Thank you so much for your critical revision. Actually the signal for 2M was present in peak which was mistakenly mentioned as 1M. We have make correction in the image. Again thanks a lot for highlighting this serious issue.

Comment: Most important: the FTIR and same biological tests and assays carried out on the nanoparticles must be performed on the extract in order to be sure monitored properties have to be ascribed to the nanoparticles and not to the extract, which could be very bio-active as well.

Response to comment: It is much valuable suggestion indeed. Our research group is also working for monitoring the properties of plant extract. The characterization and conduction of biological applications is time taking process as our lab doesn’t have enough facilities. So in current scenario we could not add the data for extract at this time.

Comment:  All the used instruments (brand and characteristics) must be added in the Materials and Methods section.

Response to comment: All the instruments (brand and characteristics) have been added and are highlighted with red color.

Comment:  Sometimes “nickle” is written instead of “nickel”.

Response to comment: Thanks a lot for pinpointing this mistake. Nickle has been replaced with nickel and is highlighted with red color.

Comment:  I don’t see agglomeration in the SEM pictures

Response to comment: In our view SEM was agglomerated. The word agglomeration has been removed from the main text.

Comment:  A curiosity: what do the author mean for nano-purewater

Response to comment: Nanopure water - also known as MilliQ - is used to make media and (most) solutions that are used to process biological samples. "Nanopure Water" is water purified using a Thermolyne Nanopure lab water system.

Comment:  Where is the TEM the authors refer to? 

Response to comment: TEM was mistakenly mentioned. We have removed it from the main manuscript.

Reviewer 3 Report

The research work is interesting. But some adjustments need to be made to improve the quality and comprehensibility of the final manuscript.

Introduction

-A few lines on the biological applications of nickel oxide nanoparticles should be included. It would be good to mention some concrete examples to know the wide spectrum of applications.

-A few lines on the main physical characteristics of nanoparticles obtained by green synthesis and the differences with nanoparticles obtained by conventional synthesis should be included.

Results

-Raman and FTIR spectra should be compared with other results. Differences in peaks and features should be mentioned and compared with spectra of nanoparticles obtained by conventional synthesis.

- From the X-ray spectra obtained it was possible to calculate the crystallinity of the ElA-NiONPs. In addition, compare the results with other reports using conventional green systems. This will allow an understanding of the changes in the structure as a function of the synthesis process.

-The biological studies (Enzyme inhibitory assay, Anticancer activity, Antileishmanial activity, Invitro antifungal, and Invitro Biocompatibility) are very interesting, but the discussion and analysis should be broadened. It would be interesting to contrast the results of the nanoparticles synthesized in this work with results on nanoparticles synthesized by other authors and their biological activity as a function of the synthesis process. All this is to highlight the advantages of green synthesis in the properties of nanoparticles. 

Author Response

Thank you so much for reviewing our article by providing valuable comments/suggestions. All comments have been properly addressed. Once again the manuscript has been critically reviewed several times for language proficiency, grammar, sentence structures, singularity and plurality of the words. All major and minor mistakes have been removed and corrected. The changes have been highlighted with red color throughout manuscript. If you provide further suggestions/recommendations, we will be very happy to address.

Reviewer 4

Comment: Raman and FTIR spectra should be compared with other results. Differences in peaks and features should be mentioned and compared with spectra of nanoparticles obtained by conventional synthesis.

Response to comment: Thanks a lot for critically revising the manuscript and making it flawless. Highly appreciated. This comment has been addressed and is highlighted with red color in results and discussion section.

Comment: From the X-ray spectra obtained it was possible to calculate the crystallinity of the ElA-NiONPs. In addition, compare the results with other reports using conventional green systems. This will allow an understanding of the changes in the structure as a function of the synthesis process.

Response to comment: Thank you so much for such a worthy comment that will for sure improve our manuscript. This comment has been addressed and crystallinity of the ElA-NiONPs has been calculated and is mentioned at the end (2.1-Characterization of ElA-NiONPs)

Comment: -The biological studies (Enzyme inhibitory assay, Anticancer activity, Antileishmanial activity, Invitro antifungal, and Invitro Biocompatibility) are very interesting, but the discussion and analysis should be broadened. It would be interesting to contrast the results of the nanoparticles synthesized in this work with results on nanoparticles synthesized by other authors and their biological activity as a function of the synthesis process. All this is to highlight the advantages of green synthesis in the properties of nanoparticles. 

Response to comment: It is much valuable suggestion indeed. Our research group is also working for monitoring the properties of plant extract. The characterization and conduction of biological applications is time taking process as our lab doesn’t have enough facilities. So in current scenario we could not add the data for extract at this time.

Reviewer 4 Report

The paper entitled “Exploring physical characterization and different bio-applications of Elagnus angustiflora orchestrated nickel oxide nanoparticles” presents the synthesis of nickel oxide NPs using Elaeagnus ansustifolia leaves extract as reducing agent (ElA-NiONPs).

I cannot recommend the publication of the paper in the current form, mainly because the characterisation of the NiONPs is not at all satisfactory. Actually the presented data do not seem correctly evaluated, and as a consequence the statements made are not right. More specific comments as follows:

1.       Line 99: if the polydispersity index obtained through DLS is 1, this means that there is a very large size dispersion, hence it is impossible that the reported value of hydrodynamic size of 21 nm is representative of the sample. To support this, the Zeta potential value of -11 mV is an index of lack of electrostatic stabilization of the dispersion, which further supports a large polydispersity/aggregation state of the dispersion. This should also be seen in the standard deviation, which instead is not presented for any of the reported value.

2.       Line 110: none of the XRD peak is assigned, a list of the peaks present in the spectrum is not sufficient, in particular if one wants to state “impurities free ElA-NiONPs”. Furthermore, I think that from such measurements the size of crystal domains could be extracted but not the size of the NPs.

3.       Line 113 and Figure 4: the average size of the NPs in Figure 4 is SURELY NOT 21 nm. If any statistical analysis was carried out, it should have been reported with the associated distribution and standard deviation. And in any case, I find it really hard to believe that the average diameter in solid state, as seen through SEM images, is EXACTLY the same as the hydrodynamic diameter measured with DLS (for the physical phenomena that are monitored with DLS, in most cases the measured size is always bigger, at maximum it can be equal to the SEM-measured diameter, but a distribution is anyway needed to confirm this). Furthermore, TEM images are not presented anywhere in the manuscript, even though they are mentioned multiple times together with the supposedly derived results.

4.       For all the above mentioned reasons the statement written between lines 201 and 204 is completely inconsistent and not supported by the data.

Other comments:

1.       Check the spelling for Nickel, which is mostly written as Nickle throughout the whole manuscript.

2.       Why are the x-axes in Figures 5 and 6 in decreasing order? It would be easier to read increasing quantities.

3.       The discussion should be largely improved, as in the current state it is mostly a summary of the data presented in the previous section.

4.       NPs sample preparation for characterisation should be described in more details.

Author Response

Thank you so much for reviewing our article by providing valuable comments/suggestions. All comments have been properly addressed. Once again the manuscript has been critically reviewed several times for language proficiency, grammar, sentence structures, singularity and plurality of the words. All major and minor mistakes have been removed and corrected. The changes have been highlighted with red color throughout manuscript. If you provide further suggestions/recommendations, we will be very happy to address.

Reviewer 5

Comment: Line 99: if the polydispersity index obtained through DLS is 1, this means that there is a very large size dispersion, hence it is impossible that the reported value of hydrodynamic size of 21 nm is representative of the sample. To support this, the Zeta potential value of -11 mV is an index of lack of electrostatic stabilization of the dispersion, which further supports a large polydispersity/aggregation state of the dispersion. This should also be seen in the standard deviation, which instead is not presented for any of the reported value.

Response to comment: Thanks a lot for critically revising the manuscript and making it flawless. Highly appreciated. The value of standard deviation is incorporated in the main manuscript.

Comment: The TEM images are not presented anywhere in the manuscript, even though they are mentioned multiple times together with the supposedly derived results.

Response to comment: TEM was mistakenly mentioned. We have removed it from the main manuscript.

Comment: Check the spelling for Nickel, which is mostly written as Nickle throughout the whole manuscript.

Response to comment: Thank you so much for improving this manuscript. The spelling of nickel has been replaced by nickel.

Comment: Why are the x-axes in Figures 5 and 6 in decreasing order? It would be easier to read increasing quantities.

Response to comment: Thanks a lot for your comment. The axis has been changed from descending to ascending order.

Comment: The discussion should be largely improved, as in the current state it is mostly a summary of the data presented in the previous section.

Response to comment: Thanks a lot for your comment that for sure has enhanced the quality of manuscript. This worthy comment has been addressed and authors have significantly worked for the improvement of this portion of manuscript and highlighted it with red color.

Comment:   NPs sample preparation for characterisation should be described in more details.

Response to comment: This worthy comment has been addressed. The details of characterizations have been added in the main manuscript and are highlighted with red color. Thanks a lot.

Round 2

Reviewer 2 Report

I'm sorry but I think the manuscript isn't suitable yet for publication in the current form. Authors, at the moment, did not address the most important issues arised by the Referees. Not only, but the word "nickle" is still present instead of nickel!

Author Response

Comment: I'm sorry but I think the manuscript isn't suitable yet for publication in the current form. Authors, at the moment, did not address the most important issues arised by the Referees. Not only, but the word "nickle" is still present instead of nickel!

Response to comment: The characterization and conduction of biological applications is time taking process as our lab doesn’t have enough facilities. So in current scenario we could not add the data for extract at this time. I hope you will understand our problem as we don’t have enough resource to extend this work further. Also, the word nickel is replaced with nickel.

Reviewer 4 Report

Unfortunately, the main points I raised (which were the most significant for the improvement of the manuscript) were not addressed at all. Only the minor comments were somehow taken care of, but I must comment on the big misunderstanding of the correction of the spelling of nickel: “nickel” is the right spelling, not nickle! The difference and the right spelling should be easily understood, given the topic of the paper. I once again cannot recommend the publication of this paper in the current form.

Author Response

Unfortunately, the main points I raised (which were the most significant for the improvement of the manuscript) were not addressed at all. Only the minor comments were somehow taken care of, but I must comment on the big misunderstanding of the correction of the spelling of nickel: “nickel” is the right spelling, not nickle! The difference and the right spelling should be easily understood, given the topic of the paper. I once again cannot recommend the publication of this paper in the current form.

Thank you so much dear sir/ madam for having quality time to improve our manuscript. Again we have done scanning electron microscopy. I hope this is fine now. At a moment we can’t perform DLS experiment due to unavailability of time and resources here in Pakistan. Also, the word nickel is replaced with nickel.